# Arpp19 Promotes Myc and Cip2a Expression and Associates with Patient Relapse in Acute Myeloid Leukemia

**DOI:** 10.3390/cancers11111774

**Published:** 2019-11-11

**Authors:** Eleonora Mäkelä, Eliisa Löyttyniemi, Urpu Salmenniemi, Otto Kauko, Taru Varila, Veli Kairisto, Maija Itälä-Remes, Jukka Westermarck

**Affiliations:** 1Turku Bioscience Centre, University of Turku and Åbo Akademi University, 20520 Turku, Finland; 2Institute of Biomedicine, University of Turku, 20520 Turku, Finland; 3Turku Doctoral Programme of Molecular Medicine, 20520 Turku, Finland; 4Department of Biostatistics, University of Turku, 20520 Turku, Finland; 5Department of Hematology, Turku University Hospital (TYKS), 20521 Turku, Finland; 6Central Laboratory, Turku University Hospital (TYKS), 20521 Turku, Finland

**Keywords:** cancer, ARPP-19, PME-1, SET, WT1, MRD

## Abstract

Disease relapse from standard chemotherapy in acute myeloid leukemia (AML) is poorly understood. The importance of protein phosphatase 2A (PP2A) as an AML tumor suppressor is emerging. Therefore, here, we examined the potential role of endogenous PP2A inhibitor proteins as biomarkers predicting AML relapse in a standard patient population by using three independent patient materials: cohort1 (*n* = 80), cohort2 (*n* = 48) and The Cancer Genome Atlas Acute Myeloid Leukemia (TCGA LAML) dataset (*n* = 160). Out of the examined PP2A inhibitors (CIP2A, SET, PME1, ARPP19 and TIPRL), expression of ARPP19 mRNA was found to be independent of the current AML risk classification. Functionally, ARPP19 promoted AML cell viability and expression of oncoproteins MYC, CDK1, and CIP2A. Clinically, ARPP19 mRNA expression was significantly lower at diagnosis (*p* = 0.035) in patients whose disease did not relapse after standard chemotherapy. ARPP19 was an independent predictor for relapse both in univariable (*p* = 0.007) and in multivariable analyses (*p* = 0.0001) and gave additive information to EVI1 expression and risk group status (additive effect, *p* = 0.005). Low ARPP19 expression was also associated with better patient outcome in the TCGA LAML cohort (*p* = 0.019). In addition, in matched patient samples from diagnosis, remission and relapse phases, ARPP19 expression was associated with disease activity (*p* = 0.034), indicating its potential usefulness as a minimal residual disease (MRD) marker. Together, these data demonstrate the oncogenic function of ARPP19 in AML and its risk group independent role in predicting AML patient relapse tendency.

## 1. Background

Acute myeloid leukemia (AML) is one of the most aggressive cancer types [1] and although up to 85% of the patients under the age of 60 achieve complete remission (CR) after standard induction therapy, only 35% to 40% can be fully cured [1,2]. In adult AML, the actual risk profile of a significant percentage of patients is not optimally reflected in current genetic classification schemes [3,4,5]. According to the European Leukemia Net risk group classification, most adult AML patients belong to the intermediate risk group [6] which practically means that they have high relapse risk after conventional chemotherapy. These patients are therefore often directed to hematopoietic stem cell transplantation (HSCT). However, all intermediate risk patients would not need HSCT but could be cured with intensive chemotherapy. Thus, together with the mortality rate up to 25% among HSCT patients and life-long need for immunosuppression with surviving patients, it would be of high clinical relevance to better understand the mechanisms that promote relapse tendency. Furthermore, although most patients in the favourable risk group can be cured with chemotherapy, some patients yet relapse. Therefore, understanding of the mechanisms behind high relapse risk would be useful to develop approaches to recognize the favorable risk group patients that would benefit from being directed to immediate HSCT. 

Protein Phosphatase 2A (PP2A) is a tumor suppressor which plays a critical role in a plethora of cancer relevant cellular processes, including regulation of cell cycle and apoptosis [7,8]. In cancer, the non-genomic inhibition of PP2A activity by elevated expression of endogenous PP2A inhibitor proteins (PAIPs), such as CIP2A, SET, PME1, ARPP19 and TIPRL, greatly exceeds the frequency of genetic mutations on PP2A genes [9]. However, while in many solid cancers, the non-genomic inhibition of PP2A has already been extensively studied, in haematological malignancies, this understanding is still relatively poor.

Due to the recent discovery of PP2A inhibition as a putative AML driver mechanism [10], and even as a potential AML therapy target [7,8,10], it is important to understand which PAIPs are clinically relevant PP2A inhibitors in AML. However, none of the published studies have systematically compared the expression profiles of different PAIPs in AML. One of the PAIPs, ARPP19 (cAMP-regulated phosphoprotein 19), a member of the alpha-endosulfine (ENSA) family, has been shown to promote G2/M transition and the mitotic state in solid cancer cells [11]. ARPP19 overexpression has been linked to tumor progression in solid cancers such as glioma [12] and hepatocellular carcinoma [13] but its role in AML has not been studied as yet.

In this first study addressing landscape of PAIPs in AML, we discovered low ARPP19 mRNA expression as a novel predictive marker for estimation of low relapse risk in patients with AML. We also identified ARPP19 as an AML oncoprotein that increases cell viability and enhances expression of oncoproteins MYC and CDK1 and also of another oncogenic PP2A inhibitor protein, CIP2A. Most importantly, we found that ARPP19 mRNA expression and its role as a predictive relapse marker was independent of current genetic risk classification schemes, suggesting that ARPP19 mediates its functions in AML by mechanisms that are independent of the known genetic mechanisms. Together, these novel results identify ARPP19 as a potential AML oncoprotein with clinical relevance. 

## 2. Materials and Methods

### 2.1. Patient Cohorts

Patient cohort1: Consecutive bone marrow samples were collected between January 2000 and July 2010, a total of 80 patients aged 18–65 diagnosed with de novo or secondary AML at Turku University Hospital (TYKS). Patients with acute promyelocytic leukemia (t(15;17)(q22;q12)) were excluded from this cohort. Patient characteristics are presented in Appendix A. Median age for the patients was 50 years (Q_1_ = 38.8, Q_3_ = 58.0), median overall survival was 5.4 years (95% CI, 2.8 to 7.9) and median follow-up time was 5.4 years (range 6 days–16 years). The ELN risk classification, based on cytogenetic and molecular findings, was used as risk stratification (Appendix A). Most patients (76) were enrolled in the Finnish Leukemia Study Group prospective protocols (Appendix A). In total, 32 patients were treated according to AML92 and 44 according to AML2003 protocol. Treatment of four patients was significantly modified due to patient-related reasons. Although patients were treated with different schedules, all received regimens based on anthracycline and high-dose cytarabine as induction therapy. High-dose cytarabine and allogenic stem cell transplantation when possible, were used as consolidation therapy. No significant differences were found between the relapse or overall survival rates of patients on the AML92 and the AML2003 treatment. Informed consent was obtained from all patients and the local Ethical Review Board of TYKS approved the study protocol. No missing data imputation was performed. 

Patient cohort2: Bone marrow samples from 48 AML patients, including nine AML patients with supplementary follow-up samples at first remission and/or at relapse, were analyzed from the Finnish Hematology Registry and Clinical Biobank (FHRB) collection. Patient characteristics for the nine patients are presented in Appendix A. All 48 patients had received intensive chemotherapy as an induction therapy and achieved CR. Additional follow-up samples at remission were available from four patients and at relapse from eight patients. Samples were collected from Finnish university hospitals and other hematological units between December 2011 and January 2017. Median age for the nine patients was 59.8 years (Q_1_ = 50.7, Q_3_ = 68.8), median overall survival was 1.7 years (95% CI, 1.3 to 3.9) and median follow-up time was 1.7 years (range 1–4.5 years). FHRB is authorized by the Finnish National Supervisory Authority for Welfare and Health (Valvira) and has been approved by the Finnish National Medical Ethics Committee. All patients signed an informed consent prior to biobanking. 

### 2.2. Statistical Analysis

Continuous variables were summarized by descriptive statistics (median, interquartile range and range) while frequencies and percentages were calculated for categorical data. Patients were stratified according to gene expression at diagnosis into high (>median expression of the studied gene in AML patients) and low (<median expression of the studied gene in AML patients). An additional analysis was performed by using an overexpression (>mean expression of the studied gene in normal sample), underexpression (<mean expression of the studied gene in normal sample) or subpopulation analysis based on the distribution profile of the studied gene expression (also including quartiles). For continuous variables, when possible, transformations (ln, sqrt) were performed to achieve a normal distribution assumption. Wilcoxon rank sum test, Kruskal–Wallis test, Student’s *t*-test and paired *t*-test were used for analyzing continuous variables. 

Frequency tables were analyzed using Fisher’s exact test for categorical variables. A Pearson’s pairwise correlation analysis was performed in a gene-to-gene manner and further hierarchical clustering (average linkage) was performed. Separate logistic regression model was fit for ARPP19 and EVI1 alone and ARPP19+EVI1 together. Discriminative power of the three models was evaluated using Receiver Operating Characteristic (ROC) curves. A chi-squared test was used for comparison of AUC-values. 

Univariable survival analysis for overall survival (OS) and time to relapse was based on the Kaplan–Meier method whereby stratum-specific outcomes were compared using log-rank statistics. To adjust for the explanatory variables (diagnosis age, risk group stratification, FLT3-ITD status, NPM1 mutation status, expression levels of CIP2A, SET, EVI1, WT1, ARPP19, TIPRL and PME1), a Cox proportional hazards regression model was used for univariable and multivariable analyses. We used a type 1 approach whereby we report the additive effect of the marker. In the multivariable analysis, covariates were entered in a stepwise backward manner. 

OS was defined for all patients measured from the date of diagnosis to the date of death from any cause. Patients not known to have died at the last follow-up were censored on the date they were last known to be alive. Time to relapse was defined for patients from the date of diagnosis until the date of relapse. Patients not known to have relapsed were censored on the date they were last examined.

### 2.3. RNA Isolation and cDNA Synthesis

Total RNA was isolated from extracted mononuclear cells (patient bone marrow samples). Total RNA was extracted using the E.Z.N.A^®^ Total RNA Kit I (Omega Bio-Tek Inc, Norcross, GA, USA) according to the manufacturer’s instructions. After isolation, RNA concentration was measured using a NanoDrop spectrophotometer (Thermo Fisher Scientific, Waltham, MA, USA). cDNA was synthesized (with 1 μg of total RNA as a starting material) using SuperScript III Reverse Transcriptase (18080093, Invitrogen, Carlsbad, CA, USA), random primers (C1181, Promega, Madison, WI, USA), RiboLock(tm) Ribonuclease Inhibitor (#EO0381, Thermo scientific, Waltham, MA, USA) and dNTP-mix (BIO-39028, Bioline, London, UK). RT-reactions were performed according to the enzyme manufacturer’s instructions. 

### 2.4. Quantitative Real-Time PCR (RQ-PCR)

Primers for each gene specific assays were designed to be located at different exonic sequences to avoid amplification of genomic DNA. The primer concentration in each reaction was 300 nM and the probe concentration was 200 nM. The specificity of RQ-PCR reactions was verified by agarose gel electrophoresis and melting curve analysis. A single band of the expected size and a single peak, respectively, were required. The amplification efficiency for each target was also assessed. shRNA control and the standard curve analysis for amplification efficiency and the melting curve analysis for ARPP19 RQ-PCR are shown in Appendix A. Amplification of target cDNAs was performed using KAPA PROBE FAST RQ-PCR Kit (Kapa Biosystems, Wilmington, MA, USA) and 7900 HT Fast Real-Time PCR System (Thermo Fisher) according to the manufacturers’ instructions. Quantitative real-time PCR was executed under the following conditions: 95 °C for 10 min followed by 45 cycles of 95 °C for 15 s and 60 °C for 1 min. Relative gene expression data were normalized to the expression level of endogenous house-keeping genes Glyceraldehyde-3-phosphate dehydrogenase (GAPDH) and beta-actin (ACTB) using the 2^-ΔΔC(t)^ method with SDS software (version 2.4.1, Applied Biosystems, Foster City, CA, USA) or with Thermo Fisher Cloud Real-time qPCR Relative Quantification application (Patient cohort2). To estimate the degree of overexpression in AML, the expression of each gene was normalized to the expression level in a commercial normal pooled (from 56 males and females) bone marrow control sample (636591, lot 1002008, Clontech Laboratories, Fremont, CA, USA). Results were derived from the average of at least two independent experiments and two technical replicates. The primer and probe sequences used in this study for RQ-PCR analysis are listed in Appendix A.

### 2.5. Cell Culture

KG-1 (ACC-14), HL-60 (ACC-3), MOLM-14 (ACC-777) and KASUMI-1 (ACC-220) cell lines were obtained from Leibniz-Institute DSMZ-German Collection of Microorganisms and Cell Cultures (Braunschweig, Germany). All the cell lines were maintained in RPMI-1640 medium (R5886, Sigma-Aldrich, Saint Louis, MO, USA) supplemented with 10% (KG-1, HL-60) or 20% (MOLM-14, KASUMI-1) heat inactivated fetal bovine serum (FBS, Gibco, Thermo Fisher Scientific), 2 mM L-glutamine (Sigma-Aldrich), 50 units/ml penicillin (Sigma-Aldrich) and 50 mg/ml streptomycin (Sigma-Aldrich). All the cell lines were routinely tested for mycoplasma contamination. 

### 2.6. Antibodies 

The following antibodies were used: rabbit polyclonal anti-ARPP19 (11678-1-AP, Proteintech Group, Rosemont, Illinois, USA), mouse monoclonal anti-cMYC (sc-40, Santa Cruz Biotechnology, Dallas, TX, USA), mouse monoclonal anti-b-actin (A1978, Sigma), mouse monoclonal anti-GAPDH (5G4 clone, HyTest, Turku, Finland), mouse monoclonal anti-cyclin D1 (sc-450, Santa Cruz), mouse monoclonal anti-CDK1 (sc-51578), and ECL HRP-linked secondary antibodies (Agilent Dako, Santa Clara, CA, USA). 

### 2.7. Western Blot Assay

Protein extracts were separated using SDS-PAGE under denaturing conditions (4–20% Mini-PROTEAN TGX Gels) and were transferred to the PVDF membrane (Bio-Rad Laboratories, Hercules, CA, USA). Membranes were blocked with 5% milk-TBST (Tris-buffered saline and 0.1% Tween 20), incubated with the indicated primary antibodies overnight at 4 °C and then, incubated ECL HRP-linked secondary antibodies at RT for 1 h. ECL Plus Western blotting reagent (GE Healthcare) was added to the membrane and film was developed. Band intensity was determined using ImageJ software (National Institutes of Health (NIH), Bethesda, MD, USA). The images of entire WB gels with protein size markers are shown in Appendix A. 

### 2.8. siRNAs, shRNAs and Cell Viability Assay

The following siRNAs were used: siGENOME Human ARPP19 (10776) siRNA (D-015338-03, Dharmacon, Lafayette, CO, USA) and siRNA Scramble (CGUACGCGGAAUACUUCGA). Transfections were performed with Nucleofector II Device (Lonza, Basel, Switzerland) and optimized programs for each cell line.

shRNA constructs were ordered as lentiviral particles from the Biomedicum Functional Genomics Unit ((FuGU), University of Helsinki, Finland) TRC1 library. ARPP19 shRNAs were TRCN0000158847 and TRCN0000160408. Control shSCR was SHC002 (Sigma). To establish the stable cell line, the ARPP19-RNAi lentivirus was transfected into HL-60 and KG1 cells with several different amounts of infectious virus. Twenty-four hours after transduction, spinoculation was performed and selection was done with puromycin at the 72 h time point. ARPP19 expression was determined through Western blot analysis and qPCR. Differences in cell viability of shARPP19 transduced cell lines compared to control shRNA cell lines were measured with CellTiter-Glo^®^ (Promega) luminescent assay (Promega) at 24 h, 48 h, 72 h, 96 h or 120 h after plating the cells. Results were derived from the average of three independent experiments.

## 3. Results

### 3.1. PP2A Inhibitor Protein mRNA Expression in AML Patient Cohort

We analyzed mRNA expression levels of PAIPs CIP2A, PME1, TIPRL, SET and ARPP19 by real-time quantitative PCR (RQ-PCR) from 80 diagnosis phase AML patients’ bone marrow (BM) samples. Patient characteristics and distribution of the patients to three clinically used risk groups (favorable *n* = 21, intermediate *n* = 37, adverse *n* = 22) based on their genetic profiles were representative of an average AML patient population (Appendix A). The representative nature of the study material was also confirmed by significant association between risk groups and overall survival (OS) of patients in this cohort (Figure 1a, *p* = 0.003 by log-rank test). Five-year survival rate was 81% for the patients in favorable (Figure 1a, blue), 51% for the patients in intermediate (red) and 27% for the patients in the adverse risk group (green). The median OS in the whole cohort was 5.4 years (95% CI, 2.8 to 7.9) and the probability of OS at five years was 52.5%. 

To estimate the degree of overexpression in AML, expression of each gene was normalized to the expression level in a pooled normal BM control sample from 56 males and females (Clontech). The degree of overexpression as well as the median expression of each target gene are shown in Appendix A. Waterfall blots of the expression patterns of the measured genes related to normal BM control set as 0 are shown in Figure 1.

As an additional indication for the representative nature of the sample material, the expression patterns of established AML markers Wilms’ tumor 1 (WT1) [14] and ectopic viral integration site-1 (EVI1) [15] were in accordance with the published literature. WT1 mRNA was overexpressed in 91% of diagnosis phase AML patients’ bone marrow as compared to normal bone marrow (Appendix A and Figure 1b), whereas EVI1 overexpression was observed in 13% (Figure 1c) of the patients. The overexpression pattern of PP2A inhibitor SET in 30% of patients (Figure 1d) was also consistent with the published literature [16]. Of the other PP2A inhibitors, TIPRL overexpression level was equal to SET (Figure 1e, 30%), whereas ARPP19 overexpression was found in 21% of patients (Figure 1f). In contrast, neither PME1 (Figure 1h, 4%), nor CIP2A (Figure 1g, 4%) were found notably overexpressed in this AML patient cohort. 

As some of the PAIPs have been previously associated to AML [10,17,18], but their relationships to each other are not clear, we used this first study addressing the landscape of PAIPs in AML to estimate their expression redundancies and mutual dependencies by Pearson’s correlation analysis. We found that PME1 levels correlated with CIP2A (Figure 1i, r = 0.52, *p* < 0.001), SET (r = 0.54, *p* < 0.001) and ARPP19 (r = 0.58, *p* < 0.001) expression. Additionally, SET expression levels correlated with TIPRL (r = 0.43, p < 0.001) and strongly with ARPP19 gene expression (r = 0.75, *p* < 0.001). Furthermore, diagnosis phase ARPP19 expression levels also correlated with WT1 (r = 0.42, *p* = 0.001) and TIPRL (r = 0.51, *p* < 0.001) gene expression. Hierarchical clustering of the correlation matrix suggests that the expression of three PP2A inhibitors, ARPP19, PME1 and SET, form a cluster with similar expression patterns across AML patient samples (Figure 1i). EVI1 gene expression did not show any significant correlation with any other target gene in this patient cohort (for all correlations *p* > 0.05).

Based on these analyses, ARPP19 is overexpressed in AML and it associates with SET that previously have been implicated in AML [17,18]. To validate the ARPP19 as a novel AML overexpressed gene in an independent patient cohort, we analysed 48 patients from the Finnish Hematology Registry and Clinical Biobank (FHRB) (cohort2) that had received intensive chemotherapy as an induction therapy. ARPP19 mRNA was overexpressed in 58% (*n* = 28) of the cohort2 sample panel (Appendix A), thus providing an independent validation for high rate of overexpression of ARPP19 in a subset of adult AML patients.

### 3.2. ARPP19 Expression Promotes AML Cell Survival

To explore the functional role of ARPP19 in AML cells, we used four established cell lines that were chosen on the basis of their diverse genetic background (DSMZ Scientific data). Consistently with patient samples at the mRNA level (Figure 1f), the Western blot analysis demonstrated variable ARPP19 protein expression levels between AML cell lines (Figure 2a,b). Interestingly, even though ARPP19 and CIP2A did not strongly correlate at the mRNA level (Figure 1i), ARPP19 protein expression correlated CIP2A protein expression in these cell lines (Figure 2a). A plausible explanation for this could be post-transcriptional stabilization of CIP2A protein in ARPP19 positive AML cells.

To address role of ARPP19 in AML cell survival, ARPP19 was down-regulated by lentiviral shRNAs in HL-60 and KG-1 cell lines expressing high endogenous ARPP19 protein levels. Indicative of pivotal role for ARPP19 in AML cell survival or proliferation, it was very challenging to maintain a long-term depletion of ARPP19 by shRNA in cell clones. However, by using early cell clones with partial ARPP19 protein knock-down, we were able to document significantly decreased cell viability in ARPP19 shRNA transduced KG-1 cells (Figure 2c,d). Partial ARPP19 inhibition also resulted in statistically significant inhibition of number of KG-1 cells in M/G2 cell cycle state (Figure 2e). Similarly to KG-1 cells, HL-60 cells were also sensitive to partial shRNA-mediated ARPP19 inhibition (Figure 2f,g). 

### 3.3. ARPP19 Promotes Expression of Oncogenic Drivers MYC, CDK1 and CIP2A in AML Cells

As a more direct support of the oncogenic role of ARPP19 in AML, transient ARPP19 knockdown by siRNA decreased expression of a well-validated PP2A target [19], and oncoprotein, MYC, and of cell cycle mediator CDK1 in both cell lines (Figure 3a,b). Very interestingly, acute depletion of ARPP19 resulted in the down-regulation of CIP2A in both cell lines, and CIP2A siRNA also inhibited ARPP19 protein levels by about 40% in both cell lines (Figure 3a,b). Down-regulation of CIP2A and of MYC upon ARPP19 silencing was validated in stably transduced HL-60 cells (Appendix A). As shown previously in other cancer models [20], CIP2A promoted MYC protein levels in both of the studied AML cell lines (Figure 3a,b). 

Together with the cell survival analysis, these results support the oncogenic role of ARPP19 in AML. The results also indicate a hierarchial co-regulation of two oncogenic PP2A inhibitors, ARPP19 and CIP2A at the protein level (Figure 3b,c). ARPP19 is known as an inhibitor of PP2A/B55 complex [11], whereas CIP2A regulates PP2A/B56 [21]. These data indicate that by means of promoting CIP2A protein expression, ARPP19 could, in principle, control both these PP2A tumor suppressor complexes (Figure 3c).

### 3.4. Low ARPP19 mRNA Expression is an Independent Predictive Relapse Marker 

To assess the potential clinical relevance of ARPP19 in human AML, we correlated expression of ARPP19 and of other mRNA markers to clinical features of the patients from cohort1. In this patent cohort, the median follow-up time of 68 AML patients who achieved complete remission (CR) was 7.2 years (range 0.2–15.9 years), and in line with a representative nature of this patient material, the patients that relapsed were more likely to be in the adverse risk group than patients who did not relapse during the follow-up time (40% vs. 14%, *p* = 0.038 by Fisher’s exact test). Notably, however, none of the PP2A inhibitor genes, including overexpressed ARPP19 (*p* > 0.05 by Kruskal–Wallis test and Appendix A), showed statistically significant association with the risk groups. On the other hand and as expected, EVI1 mRNA expression at diagnosis was significantly different between the three risk groups and its expression increased in relation to the risk group (*p* = 0.005 by Kruskal–Wallis test). Together, these results indicate that potential clinical correlations with ARPP19 are independent of genetic risk group classification of the patients. 

Importantly, supportive of the oncogenic role for ARPP19 in human AML, patients without relapse during the follow-up time had significantly lower ARPP19 expression than patients that relapsed during the follow-up time (Figure 4a, *p* = 0.035 by Wilcoxon rank-sum test). However, there was no significant difference in the rate of CR (75% vs. 88%), resistance (25% vs. 8%) or death during induction therapy (0% vs. 3%) between patients with ARPP19 underexpression or overexpression. This indicates that low ARPP19 rather associates with low relapse tendency after remission than with better induction therapy response. With regards to other evaluated mRNA markers, EVI1 was the only other marker in which expression correlated with relapse (*p* = 0.023 by Wilcoxon rank-sum test). There were no significant differences between non-relapsing and relapsing groups in any other clinical characteristics, including patient’s age, alloHSCT, secondary AML, extramedullary disease, normal karyotype, NPM1 mutation and FLT3-ITD gene fusion. Of note, most of the patients with relapse (85%) did not have FLT3-ITD nor NPM1 mutation (FLT3-ITD-, NPM1-) at diagnosis.

Kaplan–Meier estimates were used to analyze association of markers with time to relapse. As expected, the risk group of patients was a strong indicator of shorter time to relapse (*p* = 0.008 by log-rank test). Notably, patients in the lowest quartile (Q_1_) of ARPP19 expression were linked to longer relapse free time (Figure 4b, *p* = 0.029 as compared to those over lowest quartile). The five-year relapse rate was only 7% for patients with the lowest quartile expression of ARPP19, while the five-year relapse rate was 33% for patients that had ARPP19 expression higher than the lowest quartile. Importantly, directly underlining the risk group independent role for ARPP19 in relapse, patients in the lowest quartile ARPP19 expression (i.e., not relapsing patients) represented all risk groups and none of the intermediate risk group patients in this low ARPP19 cohort relapsed during >10 years follow-up time (Figure 4c). On the other hand, 27% of patients with a high relapse tendency according to higher than Q_1_ ARPP19 expression belonged to favorable risk group (Figure 4d). In addition to ARPP19, only EVI1, and SET gene expressions had any role in predicting the prevalence of relapse in this patient cohort. High EVI1 mRNA expression was a strong indicator of shorter time to relapse (Appendix A, *p* < 0.0001 by log-rank test).

### 3.5. Uni- and Multivariable Analyses

Supportive of the independent role of ARPP19 in regulating AML relapse, Cox’s univariable analysis also revealed that ARPP19 (Table 1, *p* = 0.007, HR 2.87 (95% CI, 1.33 to 6.22)), EVI1 (*p* = 0.0005, HR 1.26 (95% CI, 1.11 to 1.44)), and SET (*p* = 0.035, HR 2.36 (95% CI, 1.06 to 5.25)) expressions at the diagnosis had a significant role in predicting the time to relapse. 

Multivariable Cox’s proportional hazard model for relapse included age, FLT3-ITD status, NPM1 mutation status and mRNA expression of ARPP19, CIP2A, SET, TIPRL, PME1, EVI1 and WT1 at diagnosis. In the initial model, the significant markers for time to relapse were diagnosis age (*p* = 0.024), NPM1 mutation positivity (*p* = 0.035), EVI1 (*p* = 0.0004), SET (*p* = 0.021) and ARPP19 (*p* = 0.0008) mRNA expression. After excluding the non-significant markers, age (*p* = 0.023, HR: 1.07, 95% CI, 1.01 to 1.13), NPM1 mutation positivity (*p* = 0.048, HR: 0.031 (95% CI, 0.001 to 0.97), EVI1 (*p* = 0.0005, HR: 1.41 (95% CI, 1.16 to 1.71), SET (*p* = 0.0097, HR: 0.12 (95% CI, 0.022 to 0.59) and ARPP19 (*p* = 0.0001, HR: 58.8 (95% CI, 7.39 to 467.2) expressions were independent prognostic factors for the time to relapse (Table 2). 

Very importantly, Cox’s type1 analysis revealed that ARPP19 expression (*p* = 0.005) gave additional information in AML patients relapse prognosis after the risk group, and EVI1 mRNA expression, were depicted as significant factors in explaining the probability of relapse. Receiver operating characteristic (ROC) analysis also showed that ARPP19 together with EVI1 could be a more accurate predictor of relapse than EVI1 alone (EVI1 AUC 0.69 (95% CI, 0.48 to 0.89), ARPP19 AUC 0.67 (95% CI, 0.48 to 0.83) and ARPP19+EVI1 AUC together 0.76 (95% CI, 0.57 to 0.91); EVI1 AUC vs. EVI1+ARPP19 AUC *p* = 0.07 by Chi-Squared test, Appendix A). 

Together, these results identify low ARPP19 expression as a novel risk group independent gene associated with low relapse risk in human AML. Importantly, the predictive role of ARPP19 was additive when the currently used clinicopathological markers, including risk group classification, were taken into account. 

### 3.6. Survival Analysis Based on PP2A Inhibitor Protein mRNA Expression in AML Patients 

Next, we analyzed whether the risk group independent predictive role of ARPP19 for relapse is reflected in the overall survival of all 80 cases treated with intensive chemotherapy in cohort1. For this purpose, we used Cox’s proportional multivariable hazard model for OS, which included diagnosis age, FLT3-ITD status, NPM1 mutation status, and diagnosis phase mRNA expression levels of ARPP19, CIP2A, SET, TIPRL, PME1, EVI1 and WT1. In the initial model, the significant markers for OS were diagnosis age (*p* = 0.024) and EVI1 (*p* = 0.0127) mRNA expression. After excluding the non-significant markers, diagnosis age (Table 2, *p* = 0.0004, HR: 1.07 (95% CI, 1.03 to 1.11), NPM1 mutation positivity (*p* = 0.0165, HR: 0.21 (95% CI, 0.057 to 0.75)) and EVI1 expression (*p* = 0.0263, HR: 1.14 (95% CI, 1.02 to 1.28) were found to be independent prognostic factors for OS. Notably, out of the PAIPs, only ARPP19 mRNA expression was found as an independent prognostic factor for OS, and its HR was found to be even higher than either EVI1 or diagnosis age (*p* = 0.0456, HR: 2.05 (95% CI, 1.01 to 4.15)). 

To evaluate these results in an independent AML patient cohort, we used the RNA sequencing dataset available from The Cancer Genome Atlas (TCGA LAML, survival data available for *n* = 160, exon expression IlluminaHiSeq) [22] and analyzed the correlation between OS and ARPP19 gene expression using UCSC Xena Browser [23]. Based on median as a cut-off value, the data were categorized into two groups: low ARPP19 and high ARPP19. Consitently with other results, ARPP19 expression alone was able to act as an independent prognostic marker for OS (Figure 4e). The patient group with low ARPP19 gene expression (*n* = 82) showed better OS (*p* = 0.019 by log-rank test) than the patients with high ARPP19 expression (*n* = 78). 

In summary, the better overall survival of low ARPP19 mRNA expressing AML patients supports the observed lower relapse risk of these patients after standard therapy.

### 3.7. ARPP19 Expression Correlates with AML Disease Activity after Remission

Finally, we wanted to study whether ARPP19 expression levels correlate with the emergence of relapse after patients have achieved clinical remission. For this purpose, we could identify bone marrow samples of nine patients from cohort2 for which, in addition to diagnostic samples, follow-up samples at first remission (*n* = 4) or relapse (*n* = 8) were also available (Figure 4f). Three patients among these nine had a complete follow-up set of diagnosis, remission and relapse samples (Figure 4g). Consistently with the overexpression of ARPP19 mRNA in cohort1 (Figure 1f), seven out of nine samples in the follow-up series had higher ARPP19 mRNA expression than in the normal BM, indicated by a dashed line (Figure 4f). Notably, ARPP19 expression dropped below the control level in the remission samples, whereas it was found to be overexpressed again in the relapse samples (Figure 4f; diagnosis vs. remission *p* = 0.021, remission vs. relapse *p* = 0.034 by paired *t*-test). These findings were confirmed in the complete matched set of samples (diagnosis, remission and relapse) from three patients (Figure 4g; diagnosis vs. remission *p* = 0.047, remission vs. relapse *p* = 0.034 by a paired *t*-test). 

These results provide an independent validation for the association between high ARPP19 expression and the emergence of relapse from standard AML therapy. 

## 4. Discussion

Upon diagnosis of AML, multiple molecular markers are used to define the risk group for AML patients, but also for stratifying patients between chemotherapy and HSCT. Whereas risk grouping is sufficient to predict the relapse risk in a large fraction of patients, some patients from the favorable risk group yet relapse whereas not all intermediate-adverse risk group patients are suitable for HSCT. Therefore, a better understanding of the risk grouping independent mechanisms that affect the AML relapse tendency would be of high medical relevance. In this study, we identified ARPP19 as a novel oncogenic protein that is associated with AML relapse independently of risk groups and of other existing AML diagnostic markers. Supportive of our conclusions, ARPP19 was one of the three genes involved in the phenotypic leukemia stem cell signature which predicted poor-prognosis in the 110-subject AML cohort [24]. However, neither the independent role of ARPP19, not its risk group independent role in relapse prediction has been demonstrated in AML before our study. Collectively, these results indicate that further understanding of the mechanisms by which ARPP19 promotes relapse tendency could lead to future patient stratification strategies to quide patients with a low relapse risk to chemotherapy, whereas high relapse tendency patients (regardless of their genetic risk group) should be treated more intensively, such as with HSCT. Based on our results that ARPP19 mRNA levels faithfully followed the disease activity in patients that achieved remission with standard chemotherapy, it would be important, in the future, to evaluate the potential usefulness of ARPP19 as MRD marker which could be followed up in patients after remission to predict the emerging relapse [25]. Although mRNA expression has been considered as a challenge in MDR follow-up, recent developments in AML sample digital droplet PCR assays could make this feasible for testing in clinical trials and in clinical practice [25,26].

A decreased PP2A tumor suppressor activity due to an increased expression of PAIPs has been reported to promote the malignant growth of several cell types [9,27], including leukemic cells [10]. In AML, SET promotes both malignant growth and drug resistance [17,28], and CIP2A inhibition in AML cells reduces proliferation and MYC expression [29]. The prevalent role for PP2A inhibition in AML [10] and in other cancer types [9,27] provides a strong scientific rationale for the clinical association between low ARPP19 expression and a lower risk for AML relapse newly discovered in this study. In direct support of the oncogenic role of ARPP19 in AML, we demonstrated that ARPP19 knockdown decreased the expression of a well-validated oncogenic PP2A target MYC. Interestingly, our data also show that ARPP19 positively regulates CIP2A protein expression even though we did not observe any particularly strong assocation between ARPP19 and CIP2A mRNA expression in AML patient samples. These data suggest that similarly to CML [30,31], CIP2A may be regulated at the protein level in AML. In fact, a recent study did indicate that CIP2A protein levels function as a biomarker for AML [32]. Therefore, further studies on the regulation of CIP2A protein expression by ARPP19 in AML cells are clearly warranted. The functional hierarchy between ARPP19 and CIP2A proteins provides a plausible explanation why ARPP19 may have a stronger clinical role than CIP2A in AML. This can be rationalized as ARPP19 can control both directly its own PP2A/B55-subunit targets [33], but also PP2A/B56-subunit targets via CIP2A [21] (Figure 3c). Therapeutically, it is tempting to envision that decreased PP2A activity due to ARPP19 overexpression could be restored by blocking ARPP19 effects on PP2A. However, the development of ARPP19 targeted therapies awaits structural analysis of the ARPP19 protein.

## 5. Conclusions

In summary, our results identify ARPP19 as a potential novel AML oncoprotein. Most importantly, ARPP19 gene expression and its relapse-predicting role were found to be independent of the current genetic risk classification. This suggests that a better understanding of ARPP19 function in AML could provide clinically relevant additional value to existing diagnostic and therapeutic approaches.

## 6. Patents

J.W. and E.M have patents pending for “ARPP19 as prognostic biomarker for haematological cancers” (PCT/FI2019/050370) and “METHOD FOR PREDICTING RESPONSE TO TREATMENT WITH TYROSINE KINASE INHIBITORS” (FI20195315).

## Figures and Tables

**Figure 1 cancers-11-01774-f001:**
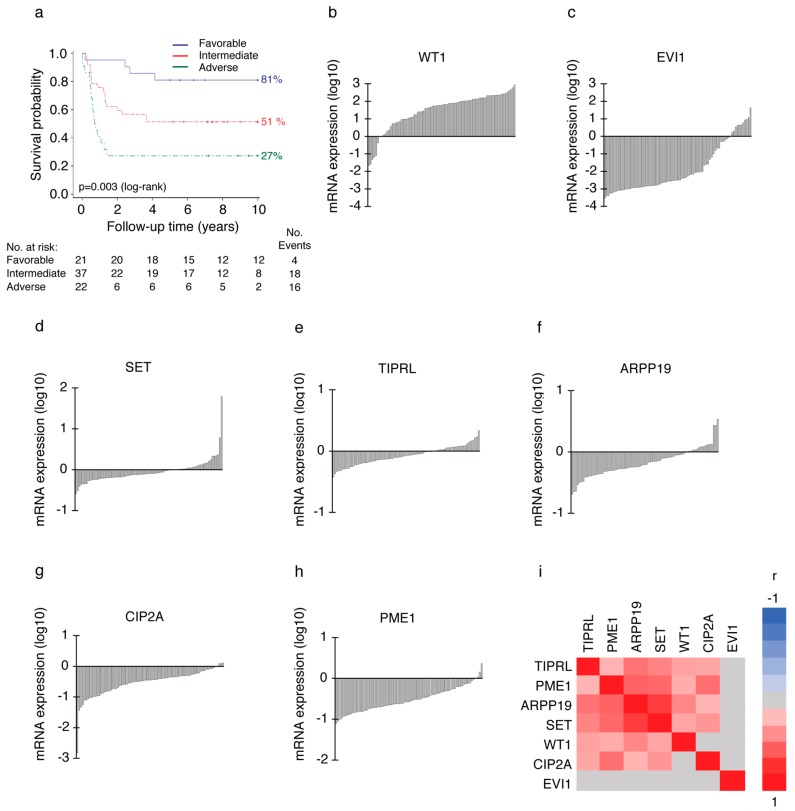
Expression profiles of PP2A inhibitors in acute myeloid leukemia (AML) patient samples. (**a**) The higher risk group is significantly associated with poor survival of AML patients in patient cohort1. *p* = 0.003 by log-rank test. Favorable *n* = 21, intermediate *n* = 37, adverse *n* = 22. (**b**) to (**h**) Waterfall blots of analysed genes from the sample panel normalized to GAPDH & b-actin expression and a pooled (*n* = 56) normal bone marrow sample. On the y-axis, log10 transformed RQ mRNA expression values derived from two technical replicates in two independent experiments. One bar represents one patient. (**b**) WT1 mRNA expression was highly overexpressed (91%) in diagnosis phase AML patients’ bone marrow compared to normal bone marrow. (**c**) EVI1 overexpression was 13%, (**d**) SET overexpression was 30%, (**e**) TIPRL overexpression was 30%, (**f**) ARPP19 overexpression was 21%, (**g**) CIP2A overexpression was 4% and (**h**) PME1 overexpression was 4% in the sample panel. (**i**) Hierarchical clustering of Pearson’s pairwise correlations for the mRNA expression of PP2A inhibitors in patient cohort1. Three potentially oncogenic PP2A inhibitors, PME1, ARPP19 and SET, form a cluster with correlated expression patterns. Red represents positive and blue negative correlation. Grey indicates non-significant correlation (*p*-value > 0.05).

**Figure 2 cancers-11-01774-f002:**
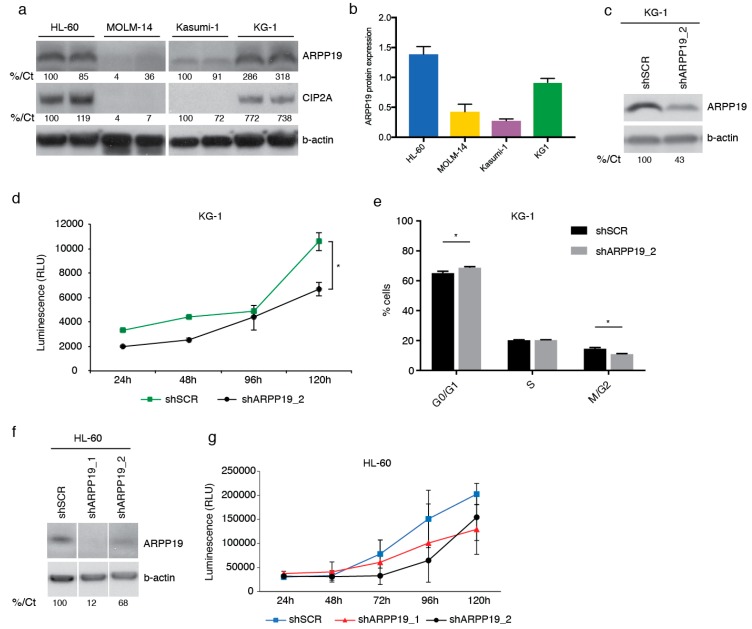
ARPP19 expression promotes AML cell survival. (**a**) Representative Western blot analysis of endogenous ARPP19 and CIP2A expression in HL-60, MOLM-14, Kasumi-1 and KG-1 cell lines. (**b**) Quantitation of ARPP19 protein levels from (**a**). B-actin was used as loading control and internal control was used to normalize the quantities on separate membranes. Internal control (T98G cell lysate) is set as 1. Shown is mean with SEM. (**c**) Representative Western blot analysis of ARPP19 expression in KG-1 cell line stably transduced with scrambled (shSCR) or ARPP19 (shARPP19_2) shRNA. (**d**) KG-1 cell viability measured with CellTiter-Glo^®^ (CTG) assay 24 h, 48 h, 96 h or 120 h after plating of stable shSCR or ARPP19 shRNA cells. Shown is mean ± SEM of three experiments. * *p* = 0.013 by Student’s *t*-test. (**e**) KG-1 cell cycle measured with propidium iodide staining and FACS analysis. Shown is mean ± SEM of two independent experiments, and significant values are indicated with asterisks (* *p* < 0.05, by 2way ANOVA). (**f**) Representative Western blot analysis of ARPP19 expression in HL-60 cell line stably transduced with scrambled (shSCR) or two different ARPP19 shRNAs (shARPP19_1 and shARPP19_2). (**g**) HL-60 cell viability measured with CTG assay 24 h, 48 h, 72 h, 96 h or 120 h after plating of stably with shSCR or two different ARPP19 shRNA transduced cells. Shown is mean ± SEM of three experiments. Numbers at the bottom of each western blot state in % the quantification of protein expression compared to control sample and normalized to b-actin expression (%/Ct).

**Figure 3 cancers-11-01774-f003:**
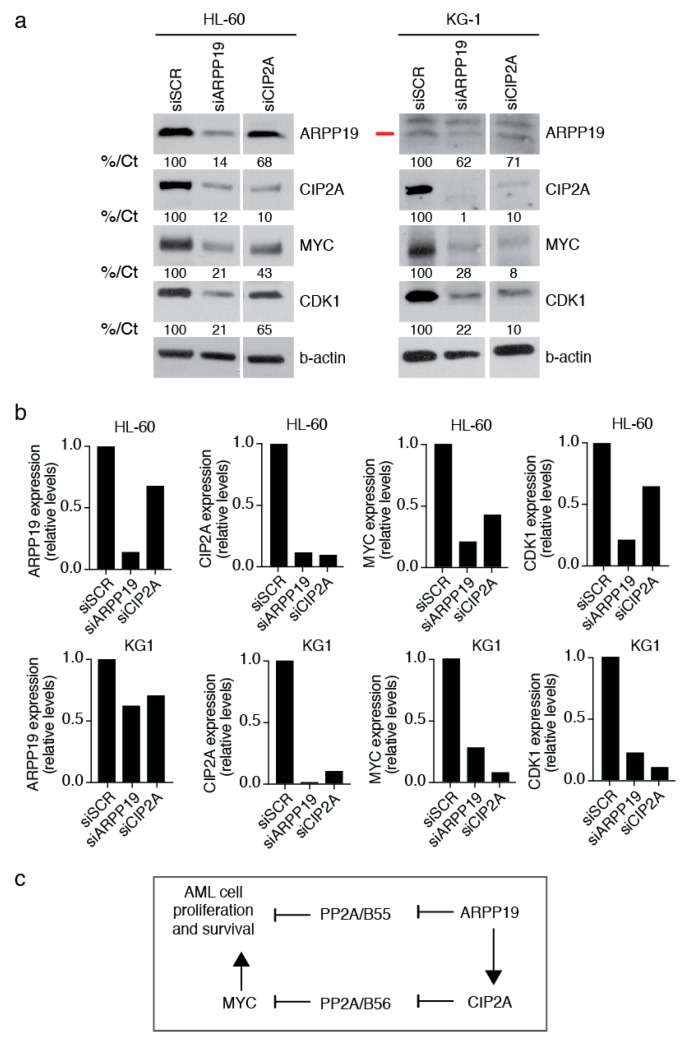
ARPP19 promotes expression of oncogenic drivers MYC, CDK1, and CIP2A in AML cells. (**a**) Western blot analysis of ARPP19, CIP2A, MYC and CDK1 expression in HL-60 and KG1 cell line 48 h after transfection with scrambled (SCR), ARPP19 (siARPP19) or CIP2A siRNA (siCIP2A). The red line marks the correct ARPP19 band. Numbers at the bottom of each western blot state in % the quantification of protein expression compared to control sample and normalized to b-actin expression (%/Ct). (**b**) Quantifications of indicated proteins normalized to b-actin from (**a**). (**c**) Schematic model of hierarchy of the two PP2A inhibitor proteins ARPP19, and CIP2A in regulation of AML cell proliferation and survival.

**Figure 4 cancers-11-01774-f004:**
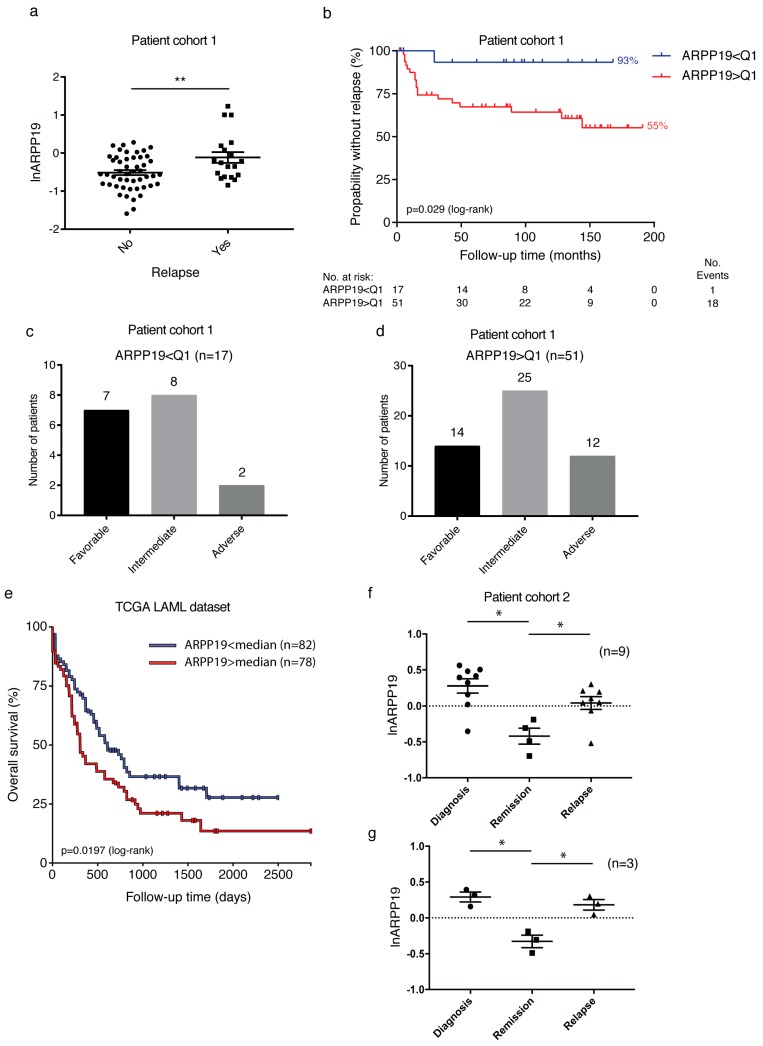
High ARPP19 mRNA expression is a risk group independent predictive relapse marker in AML. (**a**) Patients without relapse in cohort1 (*n* = 49) during the follow-up time had lower diagnostic ARPP19 mRNA expression than patients with relapse (*n* = 19). Shown is the logarithmic mean ± standard error of mean (SEM). ARPP19 expression in pooled (*n* = 56) normal bone marrow sample is set as 0. **(no vs. yes relapse) *p* = 0.0046 by Student’s *t*-test. (**b**) Lower ARPP19 expression is associated with longer time to relapse (Kaplan–Meier estimate in months) in AML patients, *p* = 0.029 by log-rank test. Q_1_ = lowest quartile of ARPP19 mRNA expression (*n* = 17). (**c**) Patients in the lowest quartile ARPP19 mRNA expression are assigned to all risk groups. (**d**) Patients over the lowest quartile ARPP19 mRNA expression are assigned to all risk groups. (**e**) Kaplan–Meier survival curve for overall survival (OS) by ARPP19 gene expression (exon IlluminaHiSeq RNAseq) in TCGA acute myeloid leukemia (LAML) patients (*n* = 160). Median serves as cut-off value. Lower ARPP19 expression is associated with longer OS in AML patients, *p* = 0.0197 by log-rank test. (**f**) ARPP19 mRNA expression in diagnosis (*n* = 9), remission (*n* = 4) and relapse (*n* = 8) samples from patients with AML. ARPP19 expression in pooled (*n* = 56) normal bone marrow sample is set as 0 (dashed line). Shown is logarithmic mean ± SEM. *(diagnosis vs. remission) *p* = 0.021, *(remission vs. relapse) *p* = 0.034 by paired *t*-test. (**g**) ARPP19 mRNA expression in matched diagnosis, remission and relapse samples from three patients with AML. *(diagnosis vs. remission) *p* = 0.047, *(remission vs. relapse) *p* = 0.034 by paired *t*-test.

**Table 1 cancers-11-01774-t001:** Univariable analysis for time to relapse in entire patient cohort1.

Parameter	Time to Relapse		
	Hazard Ratio	95% CI	*p*
Riskgroup			0.017
EVI1	1.26	1.11 to 1.44	0.0005
SET	2.36	1.06 to 5.25	0.035
ARPP19	2.87	1.33 to 6.22	0.007

**Table 2 cancers-11-01774-t002:** Multivariable analysis for time to relapse and overall survival in the entire patient cohort1.

Parameter	Time to Relapse		Overall Survival	
	Hazard Ratio	*p*	Hazard Ratio	*p*
Diagnosis age	1.07	0.023	1.07	0.0004
NPM1 mutation	0.03	0.048	0.21	0.017
EVI1	1.41	0.0005	1.14	0.026
SET	0.12	0.010		
ARPP19	58.77	0.0001	2.05	0.046

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
