# Peer review of "Arpp19 Promotes Myc and Cip2a Expression and Associates with Patient Relapse in Acute Myeloid Leukemia"

_cancers, 2019, doi:10.3390/cancers11111774_

Round 1
Reviewer 1 Report
The introduction is well written & professionally and scientifically.
Material and methods are also transparent and did not miss any information all that I could not find in the main text, I found it in the supplementary file (patients information, mutations and primer sequences).
Results and discussion are precise and the conclusion already supported by their results.
However, my primary concern that I did not completely understand how authors have chosen the PP2A for studying. The rational, and how PP2A & the associated genes selected can be addressed both in the abstract and the introduction.
Author Response
However, my primary concern that I did not completely understand how authors have chosen the PP2A for studying. The rational, and how PP2A & the associated genes selected can be addressed both in the abstract and the introduction.
Authors response: In response to this comment we have now better explained the rationale both in the abstract and in the introduction
Reviewer 2 Report
Dear Authors,
This is a great work helping to predict AML relapse based on the expression ARPP19. Experimental design is appropriate for the goal of this study. I have only one suggestion related to Western blot expression analysis. Figure 3 I suggest to quantify the expression with any of the programs available, ImageJ, i.e., and normalized based on b-actin expression. On KG-1 b-actin expression is lower in siARPP19, I was wondering if this is due to the siRNA, if so, there is no decrease on the expression. Please, check that and include on your manuscript.
Thank you very much.
Author Response
Figure 3 I suggest to quantify the expression with any of the programs available, ImageJ, i.e., and normalized based on b-actin expression. On KG-1 b-actin expression is lower in siARPP19, I was wondering if this is due to the siRNA, if so, there is no decrease on the expression. Please, check that and include on your manuscript.
Authors response: We appreciate this concern and provide now quantification of all the WB data as Fig. 3b. Quantifications revealed that actually also ARPP19 siRNA had about 40% inhibitory effect on CIP2A protein expression further highlighting interesting relationship between these two PP2A inhibitor proteins in AML cells. All the other original conclusions remained unaltered.